# Amortized Bayesian Meta-Learning for Low-Rank Adaptation of Large Language Models

## Abstract

Fine-tuning large language models (LLMs) with low-rank adaptation (LoRA) is a cost-effective way to incorporate information from a specific dataset. However, it is often unclear how well the fine-tuned LLM will generalize, i.e., how well it will perform on unseen datasets. Methods have been proposed to improve generalization by optimizing in-context prompts, or by using meta-learning to fine-tune LLMs. However, these methods are expensive in memory and computation, requiring either long-context prompts or saving copies of parameters and using second-order gradient updates. To address these challenges, we propose Amortized Bayesian Meta-Learning for LoRA (ABMLL). This method builds on amortized Bayesian meta-learning for smaller models, adapting this approach to LLMs while maintaining its computational efficiency. We reframe task-specific and global parameters in the context of LoRA and use a new hyperparameter to balance reconstruction accuracy and the fidelity of task-specific parameters to the global ones. ABMLL provides effective generalization and scales to large models such as LLAMA3-8B. Furthermore, as a result of using a Bayesian framework, ABMLL provides improved uncertainty quantification. We test ABMLL on CrossFit and Unified-QA datasets and find that it outperforms existing methods on these benchmarks in terms of both accuracy and expected calibration error.

## 1 Introduction

Large language models (LLMs) handle a variety of tasks reasonably well (Radford et al., 2019). However, to tailor LLMs to specific domains, fine-tuning on specific datasets is often necessary. While methods such as low-rank adaptation (LoRA; Hu et al., 2021) fine-tune a pretrained LLM cost-effectively, a fine-tuned LLM is limited to the domain it is trained on. Its performance may not improve in other domains and sometimes worsens as it suffers from catastrophic forgetting. Such catastrophic forgetting may result in overfitting and erasing existing capabilities of the pretrained LLM (Lazaridou et al., 2021; Luo et al., 2023).

Meta-learning is a strategy for solving this problem, training models on a variety of tasks in a way that supports generalization across tasks (Finn et al., 2017). However, meta-learning typically requires a large amount of computation and memory, making it challenging to apply to LLMs. One form of meta-learning that has been applied to LLMs involves fine-tuning models on in-context prompt-response examples (Min et al., 2022; Chen et al., 2022). Another more traditional approach, MAML-en-LLM (Sinha et al., 2024), adapts the Model-Agnostic Meta-Learning (MAML) (Finn et al., 2017) framework to LLMs. However, both methods are limited in the size of the language models that can be used: the former requires long-context prompts, whereas the latter uses second-order gradient updates and saves a model for each task.

Recent work on Amortized Bayesian Meta-Learning (ABML; Ravi & Beatson, 2019) addresses some of the computation and memory requirements of meta-learning. This approach posits a generative model over parameters where task-specific parameters are generated from global parameters, and inference over task-specific parameters is amortized. In other words, the conditional distribution over task-specific parameters is shared across tasks, implying that computation and memory costs stay constant with respect to the number of tasks. This approach thus offers a path towards efficient meta-learning for LLMs. However, several challenges exist. First, we need to specify the generative model over weight space in the context of LLMs. Second, the enormous size of LLMs makes training

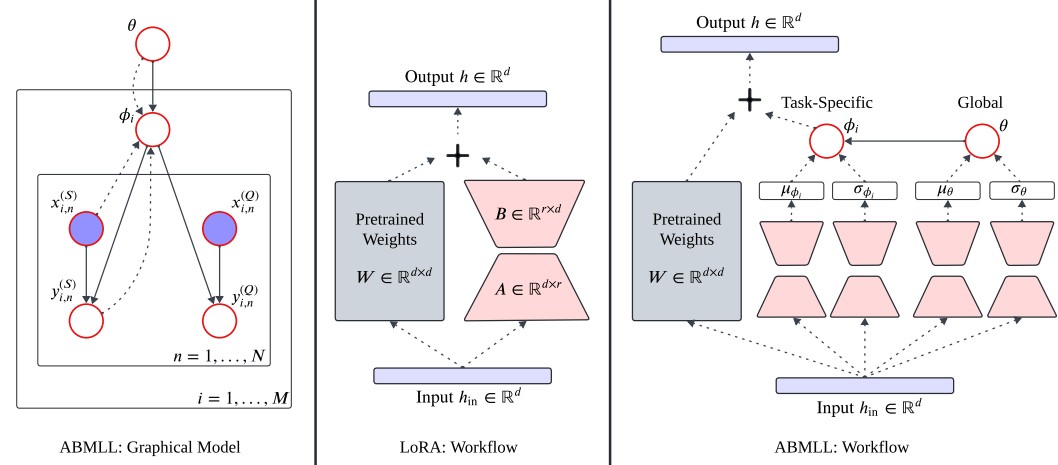

Figure 1: Illustrations of ABMLL and LoRA. There are $M$ tasks with $N$ datapoints each. $x$ is a prompt, $y$ is its output, and superscripts $S$ and $Q$ refer to the support set and the query set, which can be considered as train and test sets for individual tasks. Each solid arrow refers to a probabilistic relationship. On the graphical model shown on the left, a dashed arrow is a variational approximation; on the workflows shown to the right, a dashed arrow is an arithmetic operation.

difficult, as the scale of probabilities assigned to the model variables can overwhelm the influence of the data likelihood.

In this paper, we present a solution to these problems, taking a Bayesian approach to fine-tuning LLMs using ABML (see Figure 1). To define the underlying generative model and efficiently characterize the distributions involved, we use LoRA to express both the model weights and their uncertainty. We introduce a new prior over global variables that accounts for the spread of the parameters learned in the pretrained model. We also introduce an adjustable hyperparameter that balances reconstruction accuracy and the fidelity of task-specific parameters to global values.

Using amortized Bayesian meta-learning for LLM fine-tuning, we achieve significantly stronger performance on unseen tasks compared with regular LoRA fine-tuning. We show that amortized Bayesian meta-learning provides fine-tuned LLMs that are accurate on domain-specific tasks, more generalizable to new tasks, and provide better uncertainty estimation. Our method avoids the computation and memory overhead of other meta-learning approaches, making it adaptable to larger models such as LLAMA3-8B (Grattafiori et al., 2024) with minimal memory increase from regular LoRA. Finally, because one advantage of Bayesian methods is a natural regularization, we prune fine-tuned LLMs and show that our method is significantly stronger under pruning than both regular fine-tuning and other meta-learning methods.

## 2 RELATED WORK

**Meta-learning methods in LLMs.** Extensive work has explored meta-learning as a method for improving generalization in machine learning system, although these approaches were typically developed for models in the pre-LLM era (Finn et al., 2017; Snell et al., 2017; Ravi & Beatson, 2019; Nichol et al., 2018). Sinha et al. (2024) adapted Model-Agnostic Meta-Learning (Finn et al., 2017) to LLMs. However, this adaptation is more expensive in computation and memory than our method, requiring second-order gradient updates and saving a model for each task. More recently, Kim & Hospedales (2025) proposed a hierarchical Bayesian approach to LoRA meta-learning, but its parameters also increase linearly with number of tasks. As a result, we evaluate on larger models than the ones tried in these two papers.

In a different approach, Min et al. (2022) and Chen et al. (2022) explored meta-learning for LLMs using in-context learning. These works show that it is possible to fine-tune LLMs on in-context examples and achieve generalization. However, our approach does not require curation of such

examples, does not place constraints on the size of the context window of a model, and is more scalable.

**Uncertainty representation for LLMs.**    Approaches to capturing uncertainty for LLMs can rely on the intrinsic representation of uncertainty in the model or focus on capturing extrinsic uncertainty about model parameters. Intrinsic approaches produce better uncertainty calibration via prompt engineering and sampling (Gruver et al., 2023) or learning an external model (Shen et al., 2024). Extrinsic approaches include using fine-tuning methods to incorporate uncertainty, such as training LoRA with ensembles (Balabanov & Linander, 2024), Laplace approximation (Yang et al., 2023), and variational inference (Wang et al., 2024). Our work takes the extrinsic approach but differs from existing approaches by using meta-learning to achieve generalization across datasets.

## 3 BACKGROUND

### 3.1 LOW-RANK ADAPTATION (LoRA)

LoRA (Hu et al., 2021) fine-tunes LLM weights on a low-rank space to improve efficiency compared with regular fine-tuning. Let $\mathbf{W}_0$ of size $d_{\text{out}}$-by-$d_{\text{in}}$ denote a weight matrix from a pretrained LLM. Let $\mathbf{x}$ denote the input to $\mathbf{W}_0$, and $\mathbf{z}$ denote the output of $\mathbf{W}_0$, LoRA fine-tunes the pretrained weight $\mathbf{W}_0$ by adding a low-rank matrix comprised of two trainable matrices,

$$\mathbf{h} = (\mathbf{W}_0 + \Delta \mathbf{W}_0)\mathbf{x} = (\mathbf{W}_0 + \mathbf{B}\mathbf{A})\mathbf{x}.$$

The trainable matrices $\mathbf{B}$ and $\mathbf{A}$ are known as *LoRA adapters*. The sizes of $\mathbf{B}$ and $\mathbf{A}$ are $d_{\text{out}}$-by-$d_{\text{rank}}$ and $d_{\text{rank}}$-by-$d_{\text{in}}$, respectively, with $d_{\text{rank}}$ being significantly smaller than the original dimensions. Therefore, the number of parameters to be updated are $(d_{\text{out}} + d_{\text{in}})d_{\text{rank}}$, significantly fewer than the original $d_{\text{out}}d_{\text{in}}$.

### 3.2 APPROACHES TO META-LEARNING

Meta-learning aims to find a set of initial model parameters that can be rapidly adapted to new, unseen tasks with a few gradient steps (Schmidhuber, 1987; Bengio et al., 1991; Caruana, 1998). The strategy for doing so is to generalize from the shared statistical structure across tasks: by extracting this structure, a model can "learn to learn." A common setting in which meta-learning is used is few-shot learning, where each task might only provide a small number of examples but there are many such tasks. Personalizing large language models through modification of their weights, rather than their prompts or the context they condition on, is a natural setting for using this approach, where each user might only have a limited amount of data available but a group of users may have similar interests.

**MAML.**    A popular approach to meta-learning is the Model-Agnostic Meta-Learning (MAML; Finn et al., 2017) algorithm. This algorithm runs two training loops: an inner loop for task-specific adaptation and an outer loop for meta-optimization. Let $D_i$ denote a batch of data from task $i$, $p_\theta$ denote the prediction model parameterized by $\theta$, and $\alpha, \beta$ denote gradient descent step sizes. The goal of learning is to obtain a set of parameters $\theta_i$ for each task and a global set of parameters $\theta$ that are used to initialize learning for all tasks. In a given epoch, for each task $i$, MAML conducts an inner loop gradient update,

$$\theta_i = \theta - \alpha \nabla_\theta \mathcal{L}_{D_i}(p_\theta)$$

where $\mathcal{L}$ denotes the loss function, e.g. the cross-entropy loss. After executing a set of inner loops the outer loop update is executed,

$$\theta \leftarrow \theta - \beta \nabla_\theta \sum_i \mathcal{L}_{D_i}(p_{\theta_i'}).$$

With the outer loop update, MAML is trained to find a more generalizable set of parameters $\theta$ that is "close" to the optimal parameters for many tasks. The downside of MAML is the computational and memory requirements that can be seen in these updates. A copy of the model parameters $\theta_i$ must be cached for each task $i$. Additionally, the outer loop updates feature a gradient over the gradient of $\theta$, thus requiring a second-order gradient update.

**Reptile.** Reptile simplifies and approximates the approach to meta-learning adopted in MAML. For each task $i$, it updates the current parameters $\theta$ $k$ times, each time as a regular stochastic gradient descent,

$$\theta_i \leftarrow \theta - \alpha \nabla_\theta \mathcal{L}_{D_i}(p_\theta)$$

After these $k$ updates, a meta-update is used to improve the global parameters $\theta$,

$$\theta \leftarrow \theta + \epsilon(\theta_i - \theta),$$

with $0 < \epsilon < 1$. This can be interpreted as a gradient descent procedure where $\theta_i - \theta$ is taken as the gradient. An advantage of Reptile is efficiency: scaling with respect to the number of tasks, and not having a second-order gradient update.

### 3.3 Amortized Bayesian Meta-Learning

Amortized Bayesian Meta-Learning (ABML; Ravi & Beatson, 2019) improves upon MAML-based meta-learning frameworks by representing uncertainty with a Bayesian approach. It also amortizes inference over the parameters so that memory no longer increases linearly with the number of tasks.

Let $\theta$ denote global parameters such that a few steps of gradient descent will produce local parameters $\phi_i$ on task $i$ with dataset $D_i$. ABML treats $\theta$ as random variables, and minimizes a negative evidence lower bound using variational inference,

$$\underset{\theta}{\operatorname{argmin}} \sum_{i=1}^{M} \left[ -\mathbb{E}_{q_\theta(\phi_i|D_i)}[\log p(D_i|\phi_i)] + \text{KL}\big(q_\theta(\phi_i|D_i) \,\|\, p(\phi_i|\theta)\big) \right] + \text{KL}(q(\theta) \,\|\, p(\theta)). \quad (1)$$

The variational distribution $q_\theta(\phi_i|D_i)$ is represented by the Gaussian distribution $N(\mu_\phi, \sigma_\phi^2)$ with $\mu_\phi, \sigma_\phi$ as trainable parameters.

## 4 Method

Meta-learning enables models to develop more generalizable learning strategies. Yet, due to its computational overhead, it is underexplored on larger LLMs with billions of parameters. Our method, Amortized Bayesian Meta-Learning for LoRA (ABMLL), extends ABML, making it possible to apply to LLMs. This approach combines the advantages of meta-learning for adapting to new tasks with Bayesian probabilistic modeling for instantiating this idea and for representing uncertainty.

ABMLL uses the the objective of Eq. 1 from ABML. In our setting, $\theta$ and $\phi_i$ are the global and task-specific model parameters produced as the output of LoRA adapters. On a high level, the generative process is

$$\theta \sim p(\theta),$$
$$\phi_i \sim p(\phi_i|\theta),$$
$$D_i \sim \text{LLM}(\phi_i),$$

where $i$ represents any task $i$, and $\text{LLM}(\phi_i)$ denotes the LLM considered as a probabilistic model that takes $\phi_i$ as its inputs and outputs token sequences with joint probabilities defined by the LLM's autoregressive predictive distribution. By positing that task specific variables $\phi_i$ are generated from global variables $\theta$, the model is encouraged to learn a generalizable space of parameters with fast adaptions to different tasks. We provide pseudocode in Algorithm 1 to illustrate our approach. For any LLM layer with pretrained weights $\mathbf{W_0}$, the quantities for our extension to ABML are:

---

**Algorithm 1** One epoch in the ABMLL algorithm. The "test section" does not need to be performed every epoch.

---

**Input:** Likelihood model $p(D_i|\phi_i)$, prior $p(\theta)$ and $p(\phi|\theta)$, variational posterior $q_\theta(\phi_i|D_i)$, with trainable parameters $\mathbf{B}, \mathbf{A}$; constant $c, \beta$; number of tasks $M$ and inner-loop size $K$.

---

    **Training section**
    **for** task $i \in \{1, 2, ..., M\}$ **do**
        Inner-loop:
        **for** $k \in \{1, 2, ..., K\}$ **do**
            Draw batch $D_i$ from task $i$ dataset.
            Run a step gradient descent to minimize w.r.t. $\phi_i$,
            $-\mathbb{E}_{q_\theta(\phi_i|D_i)}[\log p(D_i|\phi_i)] + \beta\mathrm{KL}\big(q_\theta(\phi_i|D_i)\big|\big|p(\phi_i|\theta)\big).$
        **end for**
        Outer-loop:
        Run a step gradient descent to minimize w.r.t. $\theta$,
        $-\mathbb{E}_{q_\theta(\phi_i|D_i)}[\log p(D_i|\phi_i)] + \beta\mathrm{KL}\big(q_\theta(\phi_i|D_i)\big|\big|p(\phi_i|\theta)\big) + \beta\mathrm{KL}(q(\theta)||p(\theta)).$
        $\mathbf{A}_{\mu_\phi} \leftarrow \mathbf{A}_{\mu_\theta}, \quad \mathbf{A}_{\sigma_\phi} \leftarrow \mathbf{A}_{\sigma_\theta}$
        $\mathbf{B}_{\mu_\phi} \leftarrow \mathbf{B}_{\mu_\theta}, \quad \mathbf{B}_{\sigma_\phi} \leftarrow \mathbf{B}_{\sigma_\theta}$
    **end for**
    **Test section**
    Take unseen task $i$. Create a copy of the above weights, and on the new weights:
    **for** $k \in \{1, 2, ..., K\}$ **do**
        Draw batch $D_i$ from task $i$ dataset.
        Run a step gradient descent to minimize w.r.t $\phi_i$,
        $-\mathbb{E}_{q_\theta(\phi_i|D_i)}[\log p(D_i|\phi_i)] + \beta\mathrm{KL}\big(q_\theta(\phi_i|D_i)\big|\big|p(\phi_i|\theta)\big).$
    **end for**
    Evaluate on rest of data in task $i$.
    Delete the weights copy and reload the weights at the end of training section.
**Output:** $\mathbf{B}, \mathbf{A}$.

---

$$\mu_\theta = \mathbf{B}_{\mu_\theta}\mathbf{A}_{\mu_\theta},$$
$$\log\sigma_\theta^2 = \mathbf{B}_{\sigma_\theta}\mathbf{A}_{\sigma_\theta} + c\mathbf{I},$$
$$\mu_\phi = \mathbf{B}_{\mu_\phi}\mathbf{A}_{\mu_\phi},$$
$$\log\sigma_\phi^2 = \mathbf{B}_{\sigma_\phi}\mathbf{A}_{\sigma_\phi} + c\mathbf{I},$$
$$p(\phi_i|\theta) = \mathcal{N}(\phi_i; \mu_\theta + \mathbf{W_0}, \sigma_\theta^2),$$
$$q_\theta(\phi_i|D_i) = \mathcal{N}(\phi_i; \mu_\phi + \mathbf{W_0}, \sigma_\phi^2),$$
$$p(\theta) = p(\mu_\theta, \sigma_\theta) = \mathcal{N}(\mu_\theta; 0, \mathbf{I}) \cdot \mathrm{Gamma}\left(\frac{1}{\sigma_\theta^2}; a_0, b_0\right),$$
$$\mathrm{KL}(q(\theta) \,\|\, p(\theta)) = -\log p(\theta).$$

Lastly, $p(D_i|\phi_i)$ is defined as the joint probability assigned to $D_i$ where the LLM takes $\phi_i$ as its weights. The trainable parameters are the LoRA adapters $\mathbf{A}$ and $\mathbf{B}$. However, we introduce four pairs of these adapters to compute both the mean and variance of the LoRA outputs on local and global model weights. $\mathbf{I}$ is identity matrix, and $c$ is a hyperparameter constant dependent on the spread of pretrained LLM weights. $a_0$ and $b_0$ are hyperparameters, and the simplification of the KL term as $-\log p(\theta)$ follows Ravi & Beatson (2019).

**Balancing the reconstruction error.** LLMs are often overparameterized. As a result, probabilistic quantities on the space of weights, $\mathrm{KL}\big(q_\theta(\phi_i|D_i)\big|\big|p(\phi_i|\theta)\big)$ and $\mathrm{KL}(q(\theta)||p(\theta))$, can overwhelm quantities on the data space, $\log p(D_i|\phi_i)$. $\beta$-VAE (Higgins et al., 2016) and Bayesian neural network approaches (Trinh et al., 2022) introduce hyperparameters to temper the likelihood versus regularization terms. Inspired by this idea, we introduce hyperparameter $\beta$, resulting in the following objective,

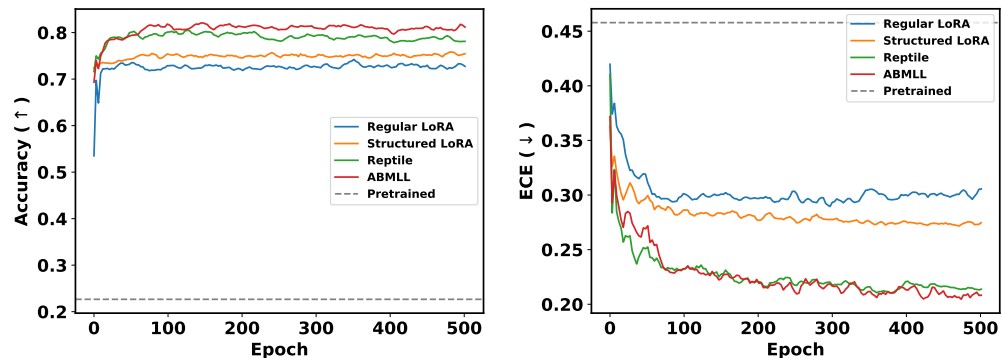

Figure 2: cls-45 validation accuracy and ECE over epochs across our method (ABMLL) and four benchmarks. Values are computed as sliding-window moving average over the three most recent epochs. ABMLL achieves consistent performance on both metrics.

$$\underset{\theta}{\operatorname{argmin}} \sum_{i=1}^{M} \left[ -\mathbb{E}_{q_\theta(\phi_i|D_i)}[\log p(D_i|\phi_i)] + \beta\mathrm{KL}\left(q_\theta(\phi_i|D_i) \,\|\, p(\phi_i|\theta)\right) \right] + \beta\mathrm{KL}(q(\theta) \,\|\, p(\theta)). \quad (2)$$

This provides a flexible way to control how close the global parameters $\theta$ are to the prior $p(\theta)$, and how close the task-specific parameters $\phi_i$ are to $\theta$.

## 5 EMPIRICAL EVALUATIONS

### 5.1 FEW-SHOT LEARNING

We first examine models fine-tuned by our ABMLL approach as few-shot learners on unseen tasks in natural text datasets.

#### 5.1.1 EXPERIMENTAL SETUP

**Model and datasets.** We fine-tune LLAMA3-8B on CrossFit (Ye et al., 2021) and UnifiedQA (Ye et al., 2021), two text datasets commonly used to train meta-learning models.

We train and evaluate in three settings. First, we use cls-45 (Ye et al., 2021), where models are trained on classification tasks, and evaluated on other distinct classification tasks. Second, we use cls-23 (Ye et al., 2021), where models are evaluated on the same classification tasks as in cls-45, but are trained on a mix of classification and other tasks including question-answering and natural language inference. Finally, following Min et al. (2022), we test stronger generalizations on more narrowly defined tasks, where models train on other tasks, but evaluate on only natural language inference (NLI), paraphrasing (Para), and knowledge-based multiple choice questions-answers (MCQA), respectively.

Because one aim of our paper is to study uncertainty quantification, we focus on multiple choice datasets. In the case of cls-45, this results in a subset of CrossFit and UnifiedQA with 34 training tasks, 15 evaluation tasks, and 68K training datapoints in total. For more details on datasets, see Section A.2 in the Appendix.

**Metrics.** We use accuracy to evaluate general performance and expected calibration error (ECE) to evaluate uncertainty estimation.

**Baselines.** We use four baseline methods that can viably scale to LLAMA3-8B. *Pretrained* is the off-the-shelf LLM. *Regular LoRA* is the default LoRA method trained on the whole randomly shuffled training dataset. *Structured LoRA* also uses the default LoRA, but the training dataset follows the same "structure" as our method: it is iteratively trained 5 gradient steps on one task at a time, dropping the global variable (and dropping the reinitialization of the task-specific variable to the

Table 1: Test accuracy and ECE across three random seeds, with standard error. Statistically significant best performances, including tied ones, are bolded.

| Method | cls-45 Acc ↑ | cls-23 Acc ↑ | NLI Acc ↑ | Para Acc ↑ | MCQA Acc ↑ |
|---|---|---|---|---|---|
| Pretrained | 26.1% ±0.1% | 26.0% ±0.1% | 57.6% ±0.0% | 57.0% ±0.0% | 71.9% ±0.0% |
| Regular LoRA | 71.6% ±0.3% | 71.4% ±0.5% | 78.5% ±0.0% | 59.9% ±0.4% | 74.8% ±0.2% |
| Struct. LoRA | 74.5% ±0.1% | 71.4% ±0.0% | 75.5% ±0.1% | 55.1% ±0.1% | 74.5% ±0.0% |
| Reptile | 73.0% ±0.3% | 72.7% ±0.2% | **83.3**% ±0.3% | **61.8**% ±0.2% | **76.2**% ±0.2% |
| ABMLL (ours) | **75.2**% ±0.0% | **73.3**% ±0.1% | 82.2% ±0.1% | **61.6**% ±1.9% | 75.9% ±0.2% |

| Method | cls-45 ECE ↓ | cls-23 ECE ↓ | NLI ECE ↓ | Para ECE ↓ | MCQA ECE ↓ |
|---|---|---|---|---|---|
| Pretrained | 0.458 ±0.000 | 0.458 ±0.000 | 0.419 ±0.000 | 0.430 ±0.000 | **0.279** ±0.000 |
| Regular LoRA | 0.318 ±0.001 | 0.328 ±0.006 | 0.310 ±0.003 | 0.433 ±0.002 | 0.302 ±0.002 |
| Struct. LoRA | 0.288 ±0.001 | 0.305 ±0.001 | 0.302 ±0.001 | 0.477 ±0.001 | 0.305 ±0.000 |
| Reptile | 0.278 ±0.000 | 0.284 ±0.001 | **0.242** ±0.004 | **0.404** ±0.003 | 0.291 ±0.002 |
| ABMLL (ours) | **0.262** ±0.001 | **0.273** ±0.005 | **0.237** ±0.020 | **0.413** ±0.007 | 0.308 ±0.003 |

global variable). Thus, it tests the effect of our generative model on performance. *Reptile* (Nichol et al., 2018) implements the Reptile meta-learning algorithm.

**Implementation details.** All experiments run 500 epochs with a single A100 GPU with 40GB of memory. All methods use the AdamW optimizer (Loshchilov & Hutter, 2017), batch-size of 2, inner-loops with 5 gradient steps, LoRA adapters with rank $= 8$ following the standard practice, and learning rate is tuned in $[10^{-6}, 10^{-4}]$. For ABMLL, $\beta = 10^{-8}$, $c = e^{-20}$. For the gamma prior, $a_0 = 1, b_0 = 0.01$ following Ravi & Beatson (2019). It takes one sample from the reparameterization step during inference. During validation on the unseen dataset, all models train 5 gradient steps on 5 batches from this dataset and evaluate on the rest.

### 5.1.2 EXPERIMENTAL RESULTS

Figure 2 shows accuracy and ECE over epochs on cls-45. We observe that the meta-learning methods (ABMLL and Reptile) achieve evidently higher accuracy than methods not based on meta-learning. Among the two meta-learning methods, the performance of ABMLL is slightly but consistently better than Reptile.

Table 1 reports test scores for each model from three random seeds, where the test results are taken from each model's best validation accuracy epoch. Statistically significant best performances, including ties, are in bold. On cls-45 and cls-23, ABMLL performs best on both metrics, suggesting that ABMLL trains a general learner able to adapt to a variety of problems. The performance of ABMLL is tied with Reptile on tasks with more distinct train-evaluation differences, possibly because the tested ability is more difficult to acquire through training tasks, bringing the performance of these two meta-learning models closer together. Meanwhile, the two meta-learning methods perform significantly better than the rest, suggesting that meta-learning is an important ingredient for fine-tuning LLMs with stronger generalization.

### 5.1.3 MEMORY CONSUMPTION

An advantage of ABMLL over many meta-learning methods is scalability. Although ABMLL introduces four pairs of LoRA adapters, pretrained weights from both ABMLL and LoRA still need to be computed during a forward pass, despite having no gradients attached to them. We compute peak memory during fine-tuning on the cls-45 dataset and find that ABMLL only requires 7.6% more memory than regular LoRA (25.6 GB for ABMLL compared to 23.8 GB for regular LoRA).

### 5.2 MODEL PRUNING

Model pruning is a way to improve the efficiency of LLMs by reducing their size and computational requirements (Ashkboos et al., 2024; Sun et al., 2023). It is also a measurement of model robustness

Table 2: Pruning results across methods on two datasets. In each column except the first, a certain percentage of neurons in each layer embedding is set to zero. ABMLL is significantly more robust against pruning than the other methods.

(a) NLI.

| Method | 0% Pruned | 1% Pruned | 10% Pruned | 20% Pruned | 30% Pruned |
|---|---|---|---|---|---|
| Pretrained | 57.6% ±0.0% | 47.9% ±0.0% | 48.2% ±0.0% | 48.2% ±0.0% | 43.6% ±0.2% |
| Regular LoRA | 78.5% ±0.0% | 69.2% ±0.4% | 68.4% ±0.3% | 66.7% ±0.5% | 60.9% ±0.6% |
| Struct. LoRA | 75.5% ±0.1% | 74.0% ±0.2% | 73.9% ±0.2% | 73.4% ±0.2% | 64.6% ±0.2% |
| Reptile | **83.3**% ±0.3% | 78.3% ±0.4% | 78.2% ±0.5% | 77.1% ±0.6% | 74.3% ±0.4% |
| ABMLL (ours) | 82.2% ±0.1% | **80.6**% ±0.4% | **80.5**% ±0.5% | **80.8**% ±0.6% | **79.0**% ±0.4% |

(b) Para.

| Method | 0% Pruned | 1% Pruned | 10% Pruned | 20% Pruned | 30% Pruned |
|---|---|---|---|---|---|
| Pretrained | 57.0% ±0.0% | 51.2% ±0.0% | 51.5% ±0.0% | 52.1% ±0.1% | 51.8% ±0.2% |
| Regular LoRA | 59.9% ±0.4% | 54.8% ±0.2% | 55.0% ±0.1% | 53.3% ±0.1% | 53.4% ±0.3% |
| Struct. LoRA | 59.9% ±0.4% | 56.0% ±0.1% | 55.5% ±0.1% | 55.1% ±0.2% | 52.9% ±0.6% |
| Reptile | **61.8**% ±0.2% | 56.1% ±0.6% | 56.0% ±0.6% | 54.8% ±0.6% | 53.2% ±0.9% |
| ABMLL (ours) | 61.6% ±1.9% | **61.1**% ±1.0% | **61.0**% ±0.9% | **60.1**% ±1.1% | **57.7**% ±1.6% |

(c) cls-45.

| Method | 0% Pruned | 1% Pruned | 10% Pruned | 20% Pruned | 30% Pruned |
|---|---|---|---|---|---|
| Pretrained | 26.1% ±0.1% | 24.4% ±0.0% | 23.0% ±0.1% | 18.8% ±0.1% | 13.5% ±0.1% |
| Regular LoRA | 71.6% ±0.4% | 65.8% ±0.9% | 65.6% ±0.9% | 65.1% ±0.7% | 61.9% ±0.8% |
| Struct. LoRA | 74.5% ±0.4% | 73.8% ±0.3% | 73.8% ±0.3% | 72.9% ±0.3% | 72.7% ±0.2% |
| Reptile | 73.0% ±0.2% | 70.8% ±0.4% | 71.0% ±0.3% | 70.8% ±0.2% | 69.6% ±0.5% |
| ABMLL (ours) | **75.2**% ±1.9% | **75.6**% ±0.2% | **75.2**% ±0.5% | **75.3**% ±0.0% | **74.3**% ±0.2% |

(d) cls-23.

| Method | 0% Pruned | 1% Pruned | 10% Pruned | 20% Pruned | 30% Pruned |
|---|---|---|---|---|---|
| Pretrained | 26.0% ±0.1% | 24.4% ±0.0% | 22.9% ±0.1% | 18.8% ±0.1% | 13.5% ±0.1% |
| Regular LoRA | 71.4% ±0.4% | 64.5% ±0.5% | 64.1% ±0.5% | 63.5% ±0.6% | 60.5% ±0.6% |
| Struct. LoRA | 71.4% ±0.4% | 71.5% ±0.3% | 71.6% ±0.3% | 71.0% ±0.2% | 70.1% ±0.4% |
| Reptile | 72.7% ±0.2% | 71.6% ±0.2% | 71.6% ±0.1% | 71.3% ±0.1% | 70.5% ±0.3% |
| ABMLL (ours) | **73.3**% ±1.9% | **72.9**% ±0.5% | **73.0**% ±0.4% | **72.5**% ±0.3% | **71.5**% ±0.4% |

as it tests model performance by removing redundant parameters, potentially dropping spurious correlations. Bayesian neural networks are known to be resource-efficient (Blundell et al., 2015). Inspired by this idea, we test the performance of ABMLL and benchmarks by setting a percentage of neurons in each layer embedding to zero, sorted by magnitudes of these neurons.

Table 2 shows that ABMLL is significantly more robust against pruning than the other methods. This result suggests that ABMLL is robust and reliable, learning generalizable features that are not as tied to specific parameters. Although regularization methods for other benchmarks such as weight-decay can potentially improve performance under pruning, we make the following observations: (1) weight-decay in general worsens performance under the no-pruning scenario, and (2) weight-decay achieves better performance than no-regularization under pruning but is still worse than ABMLL.

## 5.3 ABLATION STUDIES

In this section, we analyze the effect of different terms in our training objective, Equation 2.

Table 3 shows performance of ABMLL with different values of $\beta$ on cls-45. Results show that $\log_{10} \beta = 0$, i.e., $\beta = 1$, drops performance significantly. Thus, it is critical to balance reconstruction

Table 3: Validation accuracy and ECE across values of $\beta$ on cls-45. Lower values of $\beta$ means that the KL terms are more tempered, leading to pure maximization of data log likelihood on the objective in the limit.

| $\log_{10} \beta$ | Accuracy ↑ | ECE ↓ |
|---|---|---|
| 0 | 64.5% | 0.395 |
| -5 | 70.4% | 0.289 |
| -8 | **75.2%** | **0.262** |
| -16 | 60.6% | 0.395 |

error and the KL terms that control how close the task-specific parameters $\phi_i$ are to global parameters $\theta$, and how close $\theta$ are to the prior $p(\theta)$.

At the other extreme where the KL terms are too tempered, the objective becomes degenerate, leading to poor performance ($\log_{10} \beta = -16$). In our experiments we searched for $\log_{10} \beta \in \{0, 4, 5, 6, 7, 8, 9, 10\}$ with one random seed on cls-45 and identified the optimal value as $\log_{10} \beta = -8$. We use this setting for $\beta$ across all experiments, including Table 1.

## 6 DISCUSSION

Our results show that Bayesian modeling with meta-learning for LoRA fine-tuning achieves strong performance, measured in both accuracy and uncertainty calibration. Additionally, ablation studies demonstrate the importance of the $\beta$ hyperparameter to achieving proper balancing of the training objective. At the same time, these studies verify the validity of this objective, since tempering the KL terms too heavily leads to poor performance. In the remainder of the paper we consider the implications of these results for broader questions about Bayesian methods and inductive biases for large language models. We also identify some of the limitations of our analyses and directions for future work.

**Bridging the gap between Bayesian deep learning and LLMs.** Bayesian probabilistic modeling provides a principled way to incorporate human knowledge into models or to quantify uncertainty (Blei, 2014; Griffiths et al., 2008). In the form of Bayesian neural networks, it has been elegantly explored for smaller neural networks where uncertainty is added to their weights (MacKay, 1995; Blundell et al., 2015). The era of large models poses a challenge in connecting Bayesian methods with deep learning because of their large computational requirements. Although Bayesian methods have the potential to improve LLM interpretability, uncertainty quantification, and adaptation to new domains, they are currently under-explored (Papamarkou et al., 2024).

Our work bridges this gap by using LoRA to make LLMs amenable to Bayesian probabilistic modeling. Additionally, we use the paradigm of meta-learning to develop a generative probabilistic model that offers a novel and effective way to conduct LLM fine-tuning.

**Inductive bias in the era of LLMs.** Interpreting LLMs is challenging; incorporating inductive bias into their training or fine-tuning is even more so. *Mechanistic interpretability* is an active venue of research where LLMs are reverse-engineered to improve our understanding of their internals (Cunningham et al., 2023).

On the side of inductive bias in LLMs, one direction is meta-learning, from gradient-based approaches (Sinha et al., 2024; Kim & Hospedales, 2025) to in-context learning (Min et al., 2022; Chen et al., 2022). As a different approach, McCoy et al. (2020) constructs synthetic datasets and uses meta-learning to enforce grammatical awareness in language models prior to training on natural corpora. However, all of these methods are difficult to scale to larger models that have at least several billion parameters. Our work provides another perspective where amortized Bayesian inference captures a favorable inductive bias by modeling task-specific variables as being generated from global variables. Critically, this approach has memory overhead that is constant with respect to the number of tasks.

**Limitations and future work.** We view ABMLL as filling crucial gaps between the above areas and LLMs. However, one limitation is that its performance is comparable to the other scalable meta-

learning method, Reptile. Nevertheless, ABMLL provides consistent performance, and maintains an advantage typically cited for Bayesian neural networks which is robustness shown by retaining performance under model pruning.

Finally, there are several directions for future work. It would be valuable to test these meta-learning methods on more models and compare their effectiveness versus model sizes. Additionally, an advantage of Bayesian methods, as well as inductive bias distillation, is that they may require less data. It would be interesting to adopt ABMLL in a limited data regime and test its performance.

**Conclusion.** Meta-learning is an effective method for supporting better generalization across datasets, but its demands on computation and memory can make it difficult to apply to large language models. We have shown how meta-learning can be used to adapt LLMs by combining Amortized Bayesian Meta-Learning with Low-Rank Adaptation. This approach results in both better accuracy and stronger calibration across several benchmarks.

## REPRODUCIBILITY STATEMENT

We make sure that our results are reproducible by providing experimental details and code. Specifically, we detail our experimental setup in Section 5.1.1 and A.1 in the Appendix.

## ETHICS STATEMENT

Developing more effective methods for fine-tuning LLMs may make it easier for bad actors to train models to perform socially undesirable tasks. This is a risk shared with all methods for improving the performance of LLMs, which we hope is offset by the beneficial uses of these methods.

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

Table 4: Hyperparameters for experiments.

| Hyperparameter | ABMLL | Structured LoRA | Regular LoRA | Reptile |
|---|---|---|---|---|
| Inner loop / regular learning rate | $10^{-5}$ | $10^{-5}$ | $5 \cdot 10^{-5}$ | $10^{-5}$ |
| Outer loop learning rate | $5 \cdot 10^{-5}$ | NA | NA | NA |
| Test adaptation learning rate | $10^{-5}$ | $10^{-5}$ | $10^{-5}$ | $10^{-5}$ |
| Step size $\epsilon$ | NA | NA | NA | 0.5 |

Table 5: LoRA setup.

| | |
|---|---|
| LoRA rank | 8 |
| LoRA $\alpha$ | 16 |
| Modules using LoRA | Q Projection, V Projection, Output Projection |

## A  APPENDIX

### A.1  EXPERIMENTAL DETAILS

Here we detail the experimental setup used in our experiments. We use PyTorch with Torchtune to fine-tune LLAMA3-8B-CHAT. Each experiment uses a single A100 GPU with 40GB memory. Each experiment uses batch-size of 2 and 5 meta-learning adaptation steps. For meta-learning methods, inner loop size of 5 is used.

We detail hyperparameters in Table 4. All methods use the same LoRA setup, which is detailed in 5.

### A.2  DATASETS

We use the setup of Ye et al. (2021) for our cls-45 and cls-23 datasets (the Winogrande one uses the same training set as cls-45). We utilized the codebase provided by Min et al. (2022) to setup the datasets. Additionally, we focus on problems that can be converted into the multiple choice format. This allows us to evaluate the calibration error of models. Filtering for questions with at most four choices, we get the following training, validation, and test splits of these datasets.

cls-45 training: ['superglue-rte', 'tweet_eval-sentiment', 'glue-rte', 'superglue-wsc', 'glue-mrpc', 'tweet_eval-stance_hillary', 'tweet_eval-offensive', 'hatexplain', 'glue-cola', 'sick', 'paws', 'ethos-sexual_orientation', 'glue-qqp', 'tweet_eval-emotion', 'sms_spam', 'health_fact', 'glue-mnli', 'imdb', 'ethos-disability', 'glue-wnli', 'scitail', 'glue-sst2', 'tweet_eval-stance_abortion', 'tweet_eval-stance_climate', 'glue-qnli', 'ethos-directed_vs_generalized', 'ade_corpus_v2-classification', 'hate_speech_offensive', 'superglue-wic', 'google_wellformed_query', 'tweet_eval-irony', 'ethos-gender', 'rotten_tomatoes', 'kilt_fever']

cls-45 validation and testing: ['tweet_eval-stance_feminist', 'ethos-national_origin', 'tweet_eval-hate', 'ag_news', 'anli', 'hate_speech18', 'poem_sentiment', 'climate_fever', 'medical_questions_pairs', 'tweet_eval-stance_atheism', 'ethos-race', 'ethos-religion', 'superglue-cb', 'wiki_qa', 'yelp_polarity']

cls-23 training: ['blimp-ellipsis_n_bar_2','blimp-sentential_negation_npi_scope', 'crows_pairs', 'hellaswag', 'openbookqa', 'piqa', 'quartz-no_knowledge', 'sciq', 'ethos-disability', 'ethos-sexual_orientation', 'glue-cola', 'glue-mnli', 'glue-mrpc', 'glue-qqp', 'glue-rte', 'glue-wnli', 'hatexplain', 'health_fact', 'imdb', 'paws', 'sick', 'sms_spam', 'superglue-rte', 'superglue-wsc', 'tweet_eval-emotion', 'tweet_eval-offensive', 'tweet_eval-sentiment', 'tweet_eval-stance_hillary']

cls-23 testing: same as cls-45 testing.

As for the validation versus testing splits of other datasets, we follow the splits provided by Min et al. (2022).

We also show an example of a question from Winogrande, demonstrating the format that we use across these datasets:

Return the label of the correct answer for the question below.

Question: Jason approached Steven to deliver the official subpoena and court summons, because _ was being sued.

Choices:
A) Jason
B) Steven

