# OpenReview forum: "Amortized Bayesian Meta-Learning for Low-Rank Adaptation of Large Language Models"
_ICLR.cc/2026/Conference — Submitted to ICLR 2026_

### Official Review · Reviewer_ycgk · 2025-10-31

**Soundness:** 1
**Presentation:** 2
**Contribution:** 1
**Rating:** 2
**Confidence:** 4

**Summary:**

This paper sets out to improve the speed with which LoRA fine-tuned LLMs can generalize to unseen datasets via an amortized Bayesian meta-learning approach, with the additional benefit of providing uncertainty estimates on the predictions made by these fine-tuned LLMs. The method introduces a hierarchical Bayesian prior over LoRA weights. Each adapted weight matrix's LoRA parameters are modelled as Gaussian random variables, whose means and variances are themselves produced by LoRA adapters, yielding distributions over weight updates that encode both fit and uncertainty. Training proceeds via a $\beta$-tempered meta-ELBO, which attempts to balance data fit against a KL regularizer that ties the per-task posteriors to a global prior. The authors conduct a set of experiments on common meta-learning datasets, and show modest improvements on a couple of baselines, while remaining computationally cheaper than full MAML-style meta learning algorithms.

**Strengths:**

The paper is clearly written and does a good job of setting out the related work, the problems faced therein with the expensive 2nd order gradient calculation, and describing their proposed method.

**Weaknesses:**

Unfortunately the paper has a few weaknesses.

The first is a high-level concern about the relevance of MAML-style approaches in the age of modern pre-trained foundation models. One of the strengths of LLMs and agent systems is that they can leverage their context windows to learn how to perform a one-off task by drawing in relevant research, examples and other information to learn in-context how to solve the task. This requires no gradient updates during deployment, is cheap and simple. For tasks which will be encountered repeatedly, where the user may wish to fine-tune the model to amortize the cost of learning to perform the task, then vanilla LoRA fine-tuning is already computationally cheap, and as shown by the authors themselves in their experiments, performs well. I recommend that instead of studying MAML-style approaches in isolation, these simpler baselines which are predominantly adopted by the community should be carefully run as baselines and proven to be inferior before proceeding.

The second concern is that the method, despite the name, does not appear to be truly amortized. In Ravi & Beatson 2019, the model learns a single inference network which maps the task's dataset to the posterior parameters in one forward pass (requiring no optimization at inference time, thus yielding constant adaptation cost wrt. the number of tasks). In contrast, the method proposed in this paper takes $K$ gradient steps on each task's LoRA parameters $\phi_{i}$ at both train and test time. The amortization is therefore only structural in that the same LoRA parameterization and global prior $\theta$ is shared across tasks, however the inference itself is still optimized per task. As a result, ABMLL appears to inherit the same per-task adaptation cost as MAML/Reptile, albeit avoiding the second-order gradients.

The proposed method does not appear to be "Bayesian" either. This arises due to a number of factors; the first is that the global "posterior" $q(\theta)$ is collapsed to a point estimate due to setting $KL[q(\theta) \Vert p(\theta)] = -\log p(\theta)$, thus reducing any uncertainty arising at the meta level (i.e. if a task is far from the distribution encountered during meta-training). Further, the authors correctly note that in Bayesian deep learning, particularly for networks with large parameter counts, the regularization provided by the KL term can overwhelm the data fit and reduce performance (akin to the issues described by [Trippe & Turner, 2018](https://arxiv.org/abs/1801.06230)). While moderately tempering the posterior is seen by some practitioners as a valid way to deal with this shortcoming, the authors appear to use a very aggressive tempering of $\beta = 10^{-7}$. If the training proceeds in bf16 or fp16 mixed precision, then the contribution from any term multiplied by this value risks being numerically negligible. This, combined with the low batch size of 2 resulting in high gradient noise means that the method (as trained) effectively performs maximum-likelihood fine-tuning in LoRA space, with a tiny Gaussian prior regularizer which risks being dominated by the size of the gradient updates. For the ablation with $\beta = 10^{-16}$, the KL term is effectively zeroed out in half-precision.

Finally the evaluations in the paper are unfortunately quite limited: using only a single model - it would have been good to examine a broader range of model families and sizes (for instance spanning Llama, Qwen, Olmo, Phi, Gemma and sizes ranging from 1.5B parameters to 70B parameters). The paper makes a claim about the efficiency of this method, however the authors omit a study of the FLOPs or memory consumption of the method in comparison to baselines. The empirical performance of the method is also unfortunately slightly underwhelming, closely resembling the regular "structured" LoRA, with greater computational and memory cost as a result of the four LoRA adapters for the means and variances of the task-specific and global parameters, respectively. This additional cost and complexity does not seem to be commensurate with the performance gains.

**Questions:**

- You claim improved calibration. Have you examined whether the learned variance parameters correspond to meaningful epistemic uncertainty (e.g. via correlation with task difficulty or out-of-domain samples)?
- On the comparisons to in-context learning, in which scenarios would you expect ABMLL to outperform well-designed in-context learning baselines?
- You mention that the prior reflects the "spread" of pretrained parameters. Could you describe exactly how this was estimated? Was it layer-wise empirical variance, or simply a Normal-Gamma prior with fixed hyperparameters?
- What numeric precision was used during training? For smaller $\beta$ values of $10^{-7}$ and below, how did you ensure that these coefficients were represented accurately and did not underflow to zero in mixed-precision computation? With $\beta$ so small, and the prior's effect minimal, how do you explain the improved calibration?
- The abstract and introduction emphasised that ABMLL is computationally and memory efficient; could you provide explicit measurements (such as time per epoch, peak GPU memory, etc) compared to LoRA, Reptile, and MAML under equal settings. I also note that ABMLL introduces four pairs of LoRA adapters, roughly quadrupling adapter parameters vs vanilla LoRA. How do you control for fairness when comparing to baselines?

---

> ### Author Response · Authors · 2025-11-28
>
> We thank the reviewer for the helpful feedback, which will improve the paper.
>
> **MAML-Style Approaches in LLMs.** Thank you for the observations, and we do agree with most parts. We have two main revisions aimed at addressing these concerns.
>
> 1) We ran experiments on 3 additional datasets and show that meta-learning methods including ABMLL perform significantly better than vanilla LoRA in this few-shot learning setting. We also swept on the same hyperparameters for vanilla LoRA. Based on this, we recommend that meta-learning approaches should still be studied more in few-shot learning. The experimental details are described in the general response box and Section 5 of the revised paper.
>
> 2) In-context learning can be very effective, but ABMLL has an advantage of not requiring such in-context examples, both saving context-length and time for practitioners to tailor examples. While ABMLL quadruples computational requirements, computation also scales with sequence length squared, so ABMLL can reduce computation compared with in-context learning.
>
> **Whether ABMLL is Amortized Bayesian.** There are certain meta-learning methods (e.g. Foong, et al. (2020)) that have an additional model to map from data to weights in addition to the original model, allowing one to adapt to test tasks in one forward pass. However, Ravi & Beatson (2019) train the variational parameters for K gradient steps on a test task before evaluation, similarly to MAML which updates its model weights with K gradient steps (Algorithm 2 in the Appendix of Ravi & Beatson provides a reference). Therefore, ABMLL inherits the same per-task adaptation cost as Ravi & Beatson and Reptile.
>
> We agree with the reviewer’s statement that regularization is often tempered in Bayesian deep learning. Regarding our more aggressive tempering, LLMs have significantly more weights, and thus quantities on the weight space are larger than quantities on the data space, possibly resulting in needing a higher tempering. While the currently implemented mixed precision may or may not result in instability, we consider this an empirical issue, where we use ablation studies to find out that further tempering the KL does result in worse performance, suggesting that KL is still taken into account under the current tempering.
>
> **Variance on embeddings.** Here is the variance learned on embeddings (the h_output that is mapped to from LoRA adapters) on the last layer on two tasks: cls-45, the easier classification task where models achieve higher accuracy; paraphrasing, the harder task where all train tasks were non-paraphrase problems where models achieve lower accuracy. We observe consistently higher variance on the harder task, with up to over 30% higher variance on the down-projection MLP embeddings. The following table summarizes how much higher is the variance from paraphrasing compared to cls-45 (would be negative if lower).
>
> | Q-Projection Attention | V-Projection Attention | Out Projection Attention | Gate Projection MLP | Down Projection MLP | Up Projection MLP |
> |---|---|---|---|---|---|
> | 4.6% | 7.1% | 5.6% | 6.0% | 34.4% | 3.4% |
>
> **Additional Experiments.** Thank you for the suggestion. We ran experiments on 3 additional datasets and they show that meta-learning methods including ABMLL perform significantly better than traditional LoRA in this few-shot learning setting. We will experiment on other models in future experiments.
>
> **Computation and Memory.** We agree that the computational and memory differences result from the four instead of one pair of LoRA adapters. We checked on memory peaks, and ABMLL only requires 7.6% more memory than regular LoRA. In detail, these are: Regular LoRA - 23.8 GB; ABMLL - 25.6 GB; Reptile - 33.0 GB. During training, each method still needs to do a forward pass on the large space of pretrained weights, although these weights don't have gradients attached to them. This means ABMLL needs far less than 4 times the memory in practice. We have clarified these computational comparisons in Section 4 of the revised paper. Additionally, we note that during inference time (assuming the model no longer needs to be trained) only one pair of the adapters is required.
>
> **Clarifications.** We use the overall model empirical variance. To not to overly rely on the prior’s setup, we currently use this relatively simple approach.
>
>
>
> **References**
>
> Andrew Foong, Wessel Bruinsma, Jonathan Gordon, Yann Dubois, James Requeima, and Richard Turner. Meta-learning stationary stochastic process prediction with convolutional neural processes. Advances in Neural Information Processing Systems, 33:8284–8295, 2020.

---

### Official Review · Reviewer_NdSD · 2025-11-01

**Soundness:** 2
**Presentation:** 2
**Contribution:** 2
**Rating:** 2
**Confidence:** 3

**Summary:**

This paper combine ABMLL with LoRA to mitigates the high computational cost of Bayesian meta learning method. The results show that the proposed method achieves better performance compared to baseline methods.

**Strengths:**

* The presentation and organization of the paper are clear and easy to follow.
* The proposed method is simple and effective.
* The paper effectively mitigates the high computational cost of Bayesian meta learning through introducing LoRA.

**Weaknesses:**

* The paper essentially combines Bayesian method and LoRA in a straightforward manner. Combining Bayesian methods with LoRA is a common approach in recent work; based on this, the paper lacks furthermore novelty.

* Computational cost is a crucial aspect to the proposed method, as reducing memory usage is the primary motivation for introducing LoRA. However, the corresponding analysis is missing.

* The baseline method is relatively outdated; the authors should consider comparing with more recent approaches.

**Questions:**

* Does the proposed method require multiple samples from the posterior distribution during inference? If so, what is the additional computational cost introduced?

* According to Figure 2, ABMLL appears to require significantly more epochs for ECE to converge compared to other methods. Is this behavior consistent across all training datasets? This potential weakness should be further analyzed and discussed in the paper.

---

> ### Author Response · Authors · 2025-11-28
>
> We thank the reviewer for the helpful feedback, which will improve the paper.
>
> **Novelty.** Our work bridges several crucial gaps in the literature: applying scalable meta-learning to large models, bridging Bayesian methods and LLMs, and using amortized Bayesian meta-learning in a fine-tuning setting. For more details, please see where we discuss these points in the general response box regarding novelty. We also added more analysis of how our method can be used, where we use additional experiments to show that meta-learning is needed for few-shot learning, and ABMLL is particularly robust against model-pruning.
>
> **Discussion on Computational Cost.** Thank you for pointing this out. We checked on memory peaks, and ABMLL only requires 7.6% more memory than regular LoRA. In detail, these are: Regular LoRA - 23.8 GB; ABMLL - 25.6 GB; Reptile - 33.0 GB. During training, each method still needs to do a forward pass on the large space of pretrained weights, although these weights don't have gradients attached to them. This means ABMLL needs far less than 4 times the memory in practice. We have clarified on these computational comparisons in Section 4 of the revised paper. Meanwhile, during inference time, only one pair of LoRA adapters needs to be saved.
>
> **Baselines.** Thank you for the suggestion, and we are open to suggestions on recent scalable approaches on meta-learning. We have found and cited several recent meta-learning works on LLMs, but they require one to either save a model for each task, or use in-context learning, and these settings do not scale to our experiments (or require one to tailor in-context examples for CrossFit and UnifiedQA).
>
> **Clarification.** Only one sample is required during inference, and our results are based on this setting. We will clarify in Section 5.
>
> **Answer to question 2.** This phenomenon depends on hyperparameter choices, but for the chosen optimal hyperparameters, ABMLL sometimes converges slower, sometimes at a similar speed, depending on the dataset. However, ABMLL converges within roughly 300 gradient steps across all datasets.

---

### Official Review · Reviewer_eJXW · 2025-11-02

**Soundness:** 2
**Presentation:** 2
**Contribution:** 2
**Rating:** 2
**Confidence:** 3

**Summary:**

The paper proposes a scalable Bayesian meta-learning framework to improve the generalization and uncertainty estimation of LoRA-based fine-tuning for LLMs. Traditional meta-learning methods like MAML and Reptile are computationally expensive and memory-intensive, while LoRA fine-tuning lacks generalization across unseen domains. ABMLL adapts amortized Bayesian meta-learning by modeling global and task-specific LoRA parameters under a probabilistic framework, balancing data reconstruction with parameter regularization through a tunable β term. Experiments on CrossFit and Unified-QA benchmarks using LLAMA3-8B show that ABMLL consistently achieves higher accuracy and better calibration (lower ECE) than standard LoRA and Reptile methods, while maintaining scalability and efficiency. The approach bridges Bayesian inference with parameter-efficient LLM adaptation, offering principled uncertainty quantification and improved cross-task generalization.

**Strengths:**

+ **Reasonable integration of Bayesian meta-learning and LoRA.** The paper adapts amortized Bayesian meta-learning to large-scale LoRA fine-tuning, offering a principled probabilistic framework for task adaptation and uncertainty quantification in LLMs.
+ **Good writing.** The paper's writing is clear and easy to follow, making it accessible to the audience.
+ **Improved generalization and uncertainty calibration.** Empirical results show consistent improvements in both accuracy and expected calibration error, demonstrating enhanced generalization to unseen tasks.

**Weaknesses:**

+ **Lack of Technical Novelty.** The paper essentially merges two fairly well-known ideas: (a) PEFT via LoRA and (b) meta-learning (or Bayesian meta-learning) to obtain better generalization. One could argue that while the Bayesian angle gives a spin, the core innovation is limited. The amortised Bayesian meta-learning applied to LoRA is interesting, but meta-learning for PEFT has already been explored [1]. If the improvement comes mainly from combining existing techniques (LoRA + amortised Bayesian meta-learning), is this more an engineering/combination paper rather than a fundamentally new algorithmic contribution?
+ **Insufficient experiments: datasets and applications.** The empirical evaluation is limited to a small number of tasks/datasets (namely, Winograd and cls). While these are valid, the scope is narrow.
+ **Performance issue.** In these settings, the calibration of method does not bring significant improvement (as shown in Table. 1).
+ **Missing Baselines.** The idea of combining PEFT and Meta-Learning has been explored by prior work [1,2,3,4,5] (I might have missed some too). Please discuss the relationship between them and the reason why they are not included for comparison.

**References**
- [1] Gheini, Mozhdeh, Xuezhe Ma, and Jonathan May. "Know Where You're Going: Meta-Learning for Parameter-Efficient Fine-Tuning." arXiv preprint arXiv:2205.12453 (2022).
- [2] Hou, Zejiang, Julian Salazar, and George Polovets. "Meta-learning the difference: preparing large language models for efficient adaptation." Transactions of the Association for Computational Linguistics 10 (2022): 1249-1265.
- [3] Shao, Yihua, et al. "In-Context Meta LoRA Generation." arXiv preprint arXiv:2501.17635 (2025).
- [4] Cheng, Bo, et al. "MeTA-LoRA: Data-Efficient Multi-Task Fine-Tuning for Large Language Models." arXiv preprint arXiv:2510.11598 (2025).
- [5] Tian, Zichen, Yaoyao Liu, and Qianru Sun. "Meta-Learning Hyperparameters for Parameter Efficient Fine-Tuning." Proceedings of the Computer Vision and Pattern Recognition Conference. 2025.

**Questions:**

See above.

---

> ### Author Response · Authors · 2025-11-28
>
> We thank the reviewer for the helpful feedback, which will improve the paper.
>
> **Novelty.** Our work bridges several crucial gaps in the literature: applying scalable meta-learning to large models, bridging Bayesian methods and LLMs, and using amortized Bayesian meta-learning in a fine-tuning setting. For more details, please see where we discuss these points in the general response box regarding novelty. We view the paper as not just an algorithmic contribution because using Bayesian amortized meta-learning for LoRA requires several changes to the algorithm, including defining the space for quantities with uncertainty, tempering the objective, and choosing a sensible initialization. We then provide an analysis of how our method can be used, where we use additional experiments to show that meta-learning is needed for few-shot learning, and ABMLL is particularly robust against model-pruning.
>
> **Experiments.** Thank you for pointing this out. We conducted experiments across benchmarks on 3 additional datasets, as well as model pruning experiments. All of the new experiments showed strong performance from ABMLL, significantly outperforming traditional LoRA. The experimental details are described in the general response box and Section 5 of the revised paper.
>
> **Related work.** Thank you for pointing out these additional works. We have included more citations to this literature in the revised paper. [1], [2], [4] develop meta-learning methods for language modelling, but, similarly to MAML, save a model for each task, making memory requirements high. We instead need one model across tasks and also compare with benchmarks satisfying this feature. [5] predicts optimal hyperparameters for new tasks but differs from our focus where we learn model weights that can adapt fast. [3] is an interesting meta-learning method but needs to train another conditional variational-autoencoder and requires in-context examples that lead to long contexts.

---

### Official Review · Reviewer_Lqn3 · 2025-11-03

**Soundness:** 2
**Presentation:** 3
**Contribution:** 1
**Rating:** 2
**Confidence:** 4

**Summary:**

The paper proposes Amortized Bayesian Meta-Learning for Low-Rank Adaptation of Large Language Models (ABMLL), an approach that combines amortized Bayesian meta-learning with LoRA to enable scalable adaptation of large-scale LLMs. The method treats the global LoRA parameters as Bayesian latent variables that generate task-specific LoRA adapters, allowing the model to learn a shared probabilistic prior across tasks while efficiently adapting to new ones.

**Strengths:**

-  The paper makes an effort to address generalization as an important problem in large language model (LLM) adaptation.

 - The presentation is clear and the motivation is well established, providing a coherent link between Bayesian meta-learning and LoRA-based fine-tuning.

 - The experiments are well-presented, evaluating accuracy and expected calibration error.

**Weaknesses:**

- W1. Limited novelty over prior work.
 In Algorithm 1, the main training and testing procedures of ABMLL closely mirror Algorithm 1 and Algorithm 2 in [1], with the only modification being the reparameterization via LoRA. Since [1] also targets generalization performance in meta-learning, and the paper does not clearly articulate the conceptual or methodological advances beyond [1], the novelty of the proposed approach appears extremely limited.

 - W2. Lack of empirical comparison with strong baselines.
 Although the paper positions ABMLL as an approach for improving generalization through optimizing in-context prompts or meta-learning, it fails to empirically validate these claims against concurrent methods. In particular, the authors discuss the limitations of related work but do not provide quantitative comparisons with representative baselines such as LiFT, especially regarding peak memory cost and computational efficiency. This omission weakens the paper’s contribution relative to parallel research efforts.

 - W3. Heuristic and unstable hyperparameter choices.
 The paper acknowledges that balancing the reconstruction error is crucial, yet the corresponding hyperparameters are chosen heuristically without theoretical justification. As shown in Table 2, the method is highly sensitive to these hyperparameters, undermining robustness across tasks. Moreover, the same set of hyperparameters is used for two terms in Eq. (2), further raising concerns about its practical choice.

 - W4. Insufficient experimental motivation and limited improvement.
 The experiments mainly compare ABMLL with standard LoRA and Structured LoRA, yielding marginal performance gains. Given that the paper emphasizes cross-task generalization, the experimental design lacks motivation and diversity. It would be more convincing to include baselines that explicitly target generalization, enabling a clearer evaluation of the proposed method’s effectiveness.

[1] Ravi S, Beatson A. Amortized bayesian meta-learning. International Conference on Learning Representations. 2019.

**Questions:**

Stated in Weakness

---

> ### Author Response · Authors · 2025-11-28
>
> We thank the reviewer for the helpful feedback, which will improve the paper.
>
> **Novelty.** Our work bridges several crucial gaps in the literature: applying scalable meta-learning to large models, bridging Bayesian methods and LLMs, and using amortized Bayesian meta-learning in a fine-tuning setting. For more details, please see where we discuss these points in the general response box regarding novelty. Algorithmically, using Bayesian amortized meta-learning for LoRA requires several changes to the algorithm, including defining the space for quantities with uncertainty, tempering the objective, and choosing a sensible initialization. We will clarify this point in Section 1 of the revised paper.
>
> **Experiments and Baselines.** We conducted experiments across benchmarks on 3 additional datasets, as well as model pruning experiments. All of the new experiments showed strong performance from ABMLL, significantly outperforming traditional LoRA. The experimental details are described in the general response box and Section 5 of the revised paper.
>
> Among meta-learning methods, we compare with the similarly scalable Reptile. We have found and cited several recent meta-learning works on LLMs, but they require one to either save a model for each task (which is the case of LiFT), or use in-context learning, and these settings do not scale to our experiments.
>
> **Hyperparameter $\beta$.** While Table 2 shows very different performances across $\beta$, it is designed to use extremely wide differences in $\beta$ to sanity-check our tempered objective function. It shows that not tempering the objective at all leads to worse performances, and extreme (or degenerate) tempering also leads to bad performance. We use a simple hyperparameter tuning procedure to make sure that ABMLL is not overfitting to $\beta$: it is tuned in gaps of 10^k for different k’s, and once we found a good $\beta$ for a dataset, this $\beta$ is used for the rest of the datasets.

---

### Author Response · Authors · 2025-11-28
**General Author Response**

We thank the reviewers for their helpful feedback, which will improve the paper. We begin with a general response that clarifies the work’s novelty and summarizes newly added experiments.

**Novelty.** Although based on the existing amortized Bayesian meta-learning (ABML) technique (the reviewers do agree that the paper is clear on this aspect), ABMLL bridges several crucial gaps as a consequence of solving the problems required to apply meta-learning to large language models (LLMs). One gap is the lack of meta-learning methods that scale with the number of tasks, a critical requirement for scaling beyond smaller models and being applicable to LLMs. Although in-context learning exhibits meta-learning characteristics, we show that explicitly using meta-learning, including ABMLL, results in significant performance improvements, an aspect more clearly seen with our new experiments on additional datasets detailed below.

ABMLL also demonstrates how Bayesian methods can be applied to large foundation models by providing a scalable way to do Bayesian inference in these models.

Additionally, previous work left it unclear whether ABML can be used in a fine-tuning setting since it was developed for training a model from scratch and relies on both global and local model weights. We leverage the idea that this inductive bias might be encapsulated in pretrained weights, and that it can be extracted and improved with LoRA and a Bayesian approach. We have clarified the novelty of our work in the introduction section of the revised paper.

**Experiments on 3 additional datasets.** We have added experiments on 3 additional datasets where ABMLL has a 2% to 5% higher accuracy than regular finetuning. We also fixed an error in the implementation of cls23 from the original paper and updated Reptile with the Adam optimizer, which had been used by all other methods. The results yield several insights: 1) ABMLL clearly outperforms traditional LoRA on all datasets, and is comparable with Reptile on most datasets, with significant outperformance on one dataset; 2) meta-learning methods, namely Reptile and ABMLL, are beneficial in few-shot learning settings where both score significantly higher than traditional LoRA across all datasets.

**Experiments: model pruning.** In order to more thoroughly demonstrate the value of maintaining a posterior distribution over parameters, we investigate the effect of model pruning. ABMLL retains performance significantly better than all benchmarks under pruning, measured via the percentage of the model layer embedding units set to 0 increases (all layers). This suggests that ABMLL is especially robust and is potentially useful for not only few-shot learning, but also computational efficiency through exploring smaller model spaces. For clarification, during inference time, only one of the four pairs of LoRA adapters needs to be kept.

The details of these experiments and further discussions are included in Section 5 of the revised paper.

---

### Meta-Review · Area_Chair_66MR · 2026-01-07

**Summary:**

This paper proposes Amortized Bayesian Meta-Learning for LoRA (ABMLL), which combines amortized Bayesian meta-learning with low-rank adaptation to improve generalization and uncertainty quantification when fine-tuning large language models. The method treats global LoRA parameters as Bayesian latent variables and uses a tempered ELBO objective to balance reconstruction accuracy with parameter regularization.

While the authors made good-faith efforts to address reviewer concerns through additional experiments and clarifications, the fundamental issues of limited novelty and questionable methodological claims remain unresolved. The paper essentially combines two existing techniques without sufficient new insights, and the claimed Bayesian benefits are undermined by the aggressive tempering required to make the method work. The experimental improvements, while present, are not substantial enough to overcome these limitations.

All four reviewers consistently rated the paper as reject (Rating: 2), citing concerns about limited novelty, insufficient experimental evaluation, and questionable claims about the method's Bayesian and amortized properties.

**Reviewer Concerns:**

The primary concern across all reviews is that the contribution is essentially a straightforward combination of two well-established techniques—amortized Bayesian meta-learning and LoRA. Reviewers Lqn3 and eJXW explicitly noted that the algorithmic contribution closely mirrors prior work with only LoRA reparameterization as the modification. While the authors argue in rebuttal that bridging these methods to LLMs required addressing specific challenges (defining uncertainty spaces, tempering objectives, initialization), reviewers did not find this sufficiently novel.

**Reviewer Scores:**

reviewer will maintain their scores

---

### Decision · Program_Chairs · 2026-01-26

Reject